# Research on a Cross-Domain Few-Shot Adaptive Classification Algorithm Based on Knowledge Distillation Technology

**DOI:** 10.3390/s24061939

**Published:** 2024-03-18

**Authors:** Jiuyang Gao, Siyu Li, Wenfeng Xia, Jiuyang Yu, Yaonan Dai

**Affiliations:** 1Hubei Provincial Engineering Technology Research Center of Green Chemical Equipment, School of Mechanical and Electrical Engineering, Wuhan Institute of Technology, Wuhan 430205, China; 201615007@stu.wit.edu.cn (J.G.); 12201010010@stu.wit.edu.cn (W.X.); yjy@wit.edu.cn (J.Y.); 2School of Artificial Intelligence and Automation, Huazhong University of Science and Technology, Wuhan 430205, China; u202014939@hust.edu.cn

**Keywords:** meta-learning, few-shot, SDM, self-distillation

## Abstract

With the development of deep learning and sensors and sensor collection methods, computer vision inspection technology has developed rapidly. The deep-learning-based classification algorithm requires the acquisition of a model with superior generalization capabilities through the utilization of a substantial quantity of training samples. However, due to issues such as privacy, annotation costs, and sensor-captured images, how to make full use of limited samples has become a major challenge for practical training and deployment. Furthermore, when simulating models and transferring them to actual image scenarios, discrepancies often arise between the common training sets and the target domain (domain offset). Currently, meta-learning offers a promising solution for few-shot learning problems. However, the quantity of supporting set data on the target domain remains limited, leading to limited cross-domain learning effectiveness. To address this challenge, we have developed a self-distillation and mixing (SDM) method utilizing a Teacher–Student framework. This method effectively transfers knowledge from the source domain to the target domain by applying self-distillation techniques and mixed data augmentation, learning better image representations from relatively abundant datasets, and achieving fine-tuning in the target domain. In comparison with nine classical models, the experimental results demonstrate that the SDM method excels in terms of training time and accuracy. Furthermore, SDM effectively transfers knowledge from the source domain to the target domain, even with a limited number of target domain samples.

## 1. Introduction

At present, equipment can obtain a large amount of image data by being equipped with various sensors, and the data collected by the sensors can be used to train recognition and detection models with excellent generalization ability by means of methods such as deep learning [1]. However, in real-world scenarios, obtaining a sufficient quantity of high-quality labeled samples can be challenging due to privacy concerns, labeling costs, and other factors [2]. This often leads to the emergence of the few-shot problem. There are two primary reasons for the emergence of this issue: (1) the availability of data is adequate, but the amount of labeled data is limited, and (2) the dataset itself has a limited amount of data. In practical tests, the second reason has been found to be the primary contributor [3,4]. The few-shot problem can cause significant overfitting in deep learning models during training. Therefore, how to train a model using only a small amount of labeled data while still maintaining good generalization ability has become a critical research focus.

Few-shot learning algorithms primarily rely on data enhancement, model optimization, and transfer learning techniques [5,6]. Among these methods, transfer learning stands out as a leading research direction. This approach takes advantage of the relevance between the predicted target and the source model trained on related tasks with a large amount of data. It then applies the knowledge learned from a small number of samples to a new field. Since most few-shot tasks provide algorithms that transfer knowledge from a large collection of source datasets to a sparsely annotated collection of target categories, this method essentially falls under the category of meta-learning [7]. Therefore, this paper focuses primarily on the transfer learning algorithm based on meta-learning as its main research content.

In deep learning tasks, the training and test data are typically derived from the same dataset. However, in practical applications, the source and target data may be sampled separately, or the training and test environments may vary, leading to the cross-domain problem [8]. The cross-domain problem refers to the difference between the source domain and target domain in terms of feature space, category space, or edge distribution, which affects the model’s generalization performance in the target domain. Cross-domain problems can be categorized as domain adaptation (DA) and domain generalization (DG) problems. The primary difference between DA and DG lies in the fact that DA mainly addresses scenarios where the category spaces of the source and target domains are the same, but the edge distribution differs. Conversely, DG is designed for cases where the category spaces of the source and target domains are distinct, or there is no overlap whatsoever. In terms of complexity, DG is generally more challenging than DA. To solve the few-shot problem, it is necessary to appropriately handle the contradiction between limited data in a specific target domain and rich source domain data. Currently, two main solutions are used [9]: (1) based on supervised domain adaptation learning (SDA) methods, following the framework of domain adaptation learning, while incorporating innovative design elements to adapt to the processing requirements of few-shot datasets. (2) Based on few-shot learning techniques, enhancing the model architecture, and combining with auxiliary learning tasks (data alignment and meta-learning, etc.), to enhance the domain adaptation ability of the model.

The early SDA methods used for classification tasks relied on matrix mapping and linear classifiers, such as SHFR-ECOC and sparse feature transformation [10]. However, with the rapid development of deep learning, the emergence of end-to-end classification models has greatly simplified the complexity of feature transformation. For instance, the CCSA model [11] trains a twin network architecture through joint supervision of category entropy (i.e., softmax loss) and point-wise contrast loss, leading to fast convergence and improved classification accuracy. Another model, FADA [12], uses adversarial training to achieve feature alignment, enhancing the discriminatory power of discriminators in the network with well-designed four paired data. Another SDA method [13] uses random neighborhood embedding for domain adaptation, focusing on minimizing the maximum distance between source and target domains to achieve classification between image categories. However, the training of the aforementioned SDA methods requires a lot of time to pair samples between Ds and Dt. Although the CTL method [14] can be trained through single-stream data based on the small-batch strategy, it still needs to find a new learning paradigm to replace the tedious sample pairing work and simplify the operation process of data flow. Huang et al. [15] introduce the Aligning Distillation (ALDI) framework, which aligns students with teachers by comparing their recommendation characteristics. This alignment is achieved through tailored rating distribution alignment, ranking alignment, and identification of alignment losses, effectively narrowing the disparities between the two. Furthermore, ALDI incorporates a teacher-qualifying weighting structure to safeguard students from acquiring inaccurate information from unreliable teachers. Experimental results demonstrate that ALDI surpasses state-of-the-art baselines.

Due to the cross-domain and few-shot nature of the problem, the number of samples in the source and target domains is limited, and there are significant distribution differences between them. To address these challenges, this paper proposes a novel self-distillation mixed model (SDM). To address the issue of small sample size, this paper employs a Teacher–Student architecture and MixUp technology to perform self-distillation learning after dataset augmentation. To tackle the cross-domain problem, transfer learning is applied to learn better image representations from a relatively abundant dataset and fine-tune it in the target domain. Given the significant achievements of MixUp data augmentation technology in image classification, we use MixUp to fine-tune the model in the target domain. By mixing images between different categories, we expand the limited training dataset in the target domain and reduce the model’s memory of noisy samples to mitigate their impact on the model. In this way, the model can learn the domain information of the target domain using only a small amount of labeled data, achieving better transfer effects. Since training methods and backbones can affect the convergence speed of the model, parallel training with multiple GPUs is adopted in the experiment to increase the batch size. At the same time, ResNet is used as the backbone network of the model to improve training speed and convergence speed. Experimental results demonstrate the efficacy of both the Teacher–Student architecture and optimized MixUp technology in enhancing SDM’s classification and migration capabilities. SDM not only fully utilizes the limited labeled data to capture target domain information but also mitigates the risk of model overfitting. When compared to current mainstream models, SDM achieves the highest average classification accuracy.

## 2. Model Principle

The structure of the supervised domain-adaptive model based on self-distillation and mixing is shown in Figure 1. The primary objective of this model is to address the disparities in data distribution between the target and source domains, ultimately aiming to develop a model that demonstrates excellent generalizability within the target domain.

Model training is mainly divided into two stages: pre-training on the source domain and fine-tuning on the target domain.

(1)Pre-training stage

To enhance the learning ability of the model on the source domain data, we utilized labeled data and achieved pre-training of the SDM model (Figure 2). SDM employed a data enhancement technique that encompassed both weak and strong enhancement operations for the images in the source domain. For weak enhancements, SDM utilized methods like random cropping and flipping to improve the model’s ability to identify objects at various positions and angles. Furthermore, SDM employed a SimCLR-like strategy for strong enhancement operations, including techniques such as random color changes and Gaussian blurring, to enhance the model’s robustness to variations in illumination, color, and texture. During training, SDM fed the weakly enhanced images into the Teacher model and the strongly enhanced images into the Student model to obtain their intermediate feature and outcome vectors. Subsequently, SDM optimized the model parameters by calculating the classification loss of the Student model and the alignment loss of the Teacher with the Student model. By adjusting the variable parameters, SDM was able to flexibly control the weights of the two losses to balance classification accuracy and model consistency.

In contrast to the previous self-distillation method that employed pseudo-twin networks for self-distillation using two identical models, SDM utilized the Teacher–Student structure for supervised self-distillation training. This approach encompassed two distinct components: supervised classification training and aligned distillation training. Both the Teacher and Student models employed an identical model based on the convolutional section of the ResNet50 architecture, which had been pre-trained on ImageNet. Additionally, two fully connected layers were appended to each model. During training, the Teacher received slightly enhanced images as input, whereas the Student received reinforced images. The Teacher model was updated using a gradient and an exponential moving average (EMA) [16]. This approach can be formally expressed as follows [17]:$$W_{\mathrm{Teacher}}^{t}=(1-\beta )W_{\mathrm{Student}}^{t}+\beta W_{\mathrm{Teacher}}^{t-1}\;\beta =0.9$$

To begin, the output of the model lacks reliability due to the initialization of the fully connected layer. Therefore, during the initial number of training rounds, SDM solely employed the cross-entropy loss function between the output results of the Student model and the corresponding labels. However, as training progressed, SDM expanded the loss function to encompass both the cross-entropy loss and the cosine loss between the output features of the Student model and the Teacher model. This expanded loss function can be formulated as follows:$$L=\left\{\begin{array}{c} \mathrm{CE}({\hat{y}}_{\mathrm{Student}}\;,\;y),\;t\leq num\\ \mathrm{CE}({\hat{y}}_{\mathrm{Student}}\;,\;y)+\alpha_{t}(1-cos({\hat{x}}_{\mathrm{Student}}\;,\;{\hat{x}}_{\mathrm{Teacher}})),\;t>num \end{array}\right.$$
where the parameters of the coordination loss are related to the training rounds, which can be expressed as:$$\alpha_{t}=\alpha_{T}\times \frac{t}{T}$$

To address the issue of random domain drift between the strongly enhanced images and the original domain images, SDM employed the cosine alignment loss to ensure that the model maps images with strong and weak enhancements into similar spaces. This adaptation allows the model to adapt to different domains, resulting in improved generalization performance. Additionally, using an updated Teacher model through exponential moving average provides stability and resistance to noise interference, leading to more stable parameter learning. Therefore, SDM opted to utilize the Teacher model obtained from source domain training to ensure both model stability and accuracy.

(2)Fine-tuning stage

To further enhance the labeled images in the target domain, we fine-tuned the SDM using the pre-trained Teacher model. This process involves an image hybridization operation, as depicted in Figure 3. To create hybrid images, SDM randomly selects two images from the target domain and combines them using parameters from the Beta distribution. These hybrids are then used alongside their corresponding labels to compute cross-entropy losses. These losses are then utilized to update the model. This hybrid approach is designed to enhance the model’s ability to capture specific feature representations of the target domain, thereby improving its performance within that domain.

For two randomly selected labeled pictures in the target domain, mix the pictures and labels in a certain proportion, which can be expressed as:$$\left\{\begin{array}{c} \tilde{x}=\lambda x_{i}+(1-\lambda )x_{j}\\ \tilde{y}=\lambda y_{i}+(1-\lambda )y_{j} \end{array}\right.$$
where the mixture parameter is randomly generated, conforming to the Beta distribution,
$$f(x;a,b)=\frac{\Gamma (a+b)}{\Gamma (a)\Gamma (b)}x^{a-1}{\left(1-x\right)}^{b-1}$$

To ensure that the model can effectively identify “unseen” target domain data, it is crucial for SDM to balance its learning difficulty. However, using MixUp fully can make the model difficult to learn. Therefore, SDM has devised a random MixUp strategy to address this issue. This strategy can be summarized as follows:$$L=\left\{\begin{array}{c} \mathrm{CE}(\mathrm{model}(x_{i})\;,\;y_{i}),\;70\%\\ \lambda \mathrm{CE}(\mathrm{model}(\tilde{x})\;,\;y_{i})+(1-\lambda )\mathrm{CE}(\mathrm{model}(\tilde{x})\;,\;y_{j}),\;30\% \end{array}\right.$$

The MixUp pseudocode is as follows:

#*y*1, *y*2 should be one-hot vectors

For (*x*1, *y*1), (*x*2, *y*2) in zip (loader1, loader2):

Lam = numpy.random.beta(alpha, alpha)

*x* = Variable (lam × *x*1 + (l. − lam) × *x*2)

*y* = Variable(lam × *y*l + (l. − am) × *y*2)

optimizer. zero_grad()

Loss (net(*x*), *y*). backward ()

optimizer. step()

Self-distillation, as an important learning technique, allows the model to learn from itself, improving prediction accuracy by enhancing the model’s generalization ability and reducing the occurrence of overfitting. This method has been effectively proven in various research studies [17,18,19]. On the other hand, mixed data augmentation effectively enriches the diversity of training data by combining non-traditional input and target example transformations. This increased diversity is crucial for building more resilient and generalized models. The integration of these two strategies into SDM methods is expected to form a more robust modeling solution and improve the model’s ability to defend against adversarial attacks.

In summary, the domain-adaptive model, which is based on self-distillation and mixing, undergoes both pre-training and fine-tuning. SDM incorporates enhanced data augmentation and image-mixing strategies to enhance its generalization and adaptability to the target domain.

## 3. Experiments

### 3.1. Experimental Setup

(1)Experimental environment setting

The paper mentions using Python2.7 as the software environment, with a hardware environment of two NVIDIA2080Ti GPUs (NVIDIA, Santa Clara, CA, USA). The distribution function of the torch library is used for parallel training, so that more images can be computed in one iteration, which is helpful for the convergence of the model.

The operating system platform was Ubuntu20.04 with support for the SFAN (Serverless Auto-Scale) function. Ubuntu SFAN is used for parallel training, which is only used to facilitate validation on larger datasets with more categories.

(2)Selection of the dataset

The Office-31 dataset [20] and the OfficeHome dataset [21] are currently the most widely used small-sample cross-domain test datasets. The Office-31 dataset (Figure 4) comprises 31 common object categories, distributed across three distinct data domains: amazon, DSLR, and webcam. The sample size varies among these domains, with each class ranging from a dozen to a hundred images. The Office-31 dataset, therefore, exemplifies the typical challenges posed by small sample sizes across domains. Notably, the amazon domain is the most extensive, containing 2817 images, while the webcam and DSLR domains are more compact, with 795 and 498 images, respectively. The OfficeHome dataset comprises images from four distinct fields: art images, clip art, product images, and real-world images. Within each domain, the dataset encompasses images of 65 object categories. The OfficeHome dataset is also representative of cross-domain small-sample datasets.

To address the issue of small sample size across domains, it is necessary to perform image classification with one domain serving as the source domain and the other as the target domain. This includes comparisons such as amazon > DSLR, amazon > webcam, webcam > DSLR, webcam > amazon, DSLR > amazon, DSLR > webcam. Classifying images from the amazon domain involves semi-natural composite images of real objects against a pure white background. Webcam images are those of real objects captured in real-life settings using a web camera, while DSLR images are high-definition images of real objects taken in real-life settings using a digital SLR camera. The amazon domain exhibits distribution differences due to its non-real background, and there are also visual disparities in imaging quality between webcam and DSLR images.

To address the domain adaptation challenge posed by distribution differences, this paper aims to demonstrate that SDM outperforms other state-of-the-art models in terms of classification accuracy. SDM is tested on Office31 and several other datasets that follow the standard protocol for supervised domain adaptation. 

For the Office31 dataset, we randomly selected 20 images from the amazon domain as the source domain, 8 images from the DSLR and webcam domains served as the training set, 3 images from the target domain were randomly chosen for fine-tuning with labels, and the remaining images were used for testing. 

For the OfficeHome dataset, all source domain data were used for training. Three images were randomly selected from the target domain as labeled data, and the remaining images served as the test set. To ensure a fair comparison, the model framework was modified to Alexnet [22,23,24], as all other models used this framework on the OfficeHome dataset.

(3)Adjustment of the dataset

In the pre-training phase, it is essential to utilize both weak and strong enhancement techniques for model training. Here are the specifics:(1)Weak Image Enhancement: The initial pre-processing stage involves randomly cropping the images into smaller patches and horizontally flipping them. The patches are then resized to match the input dimensions of ResNet50, typically 224 × 224 pixels. Subsequently, these patches are converted into Tensors and normalized for further processing.(2)Strong Image Enhancement: SDM employs the MoCo v2 enhancement technique, which randomly crops the image to 224 × 224 pixels. The cropping ratios vary between 20% and 100% of the original image size, ensuring that the model is exposed to diverse image perspectives. Additionally, SDM randomly applies color jitter transformation with parameters (0.4, 0.4, 0.4, 0.1). These parameters govern the ranges of brightness, contrast, saturation changes, and stochasticity strength, respectively. There’s an 80% chance of applying this transformation to the image. Furthermore, SDM has a 20% probability of converting the image to a grayscale version and a 50% chance of applying Gaussian blur. Finally, the image is resized to the standard input dimensions for ResNet50, transformed into a Tensor, and normalized.

The phase of fine-tuning in the target domain employs the identical weak enhancement technique as the pre-training phase. During the test phase, only the image size was scaled and normalized.

(4)The SDM model settings

(1) Modify the Resnet50 of the classification head

Since the Resnet50 in Pytorch is trained on ImageNet, its classification head output is a 1000-dimensional vector, which is different from the actual category dimensions of Office31 and OfficeHome. Therefore, its classification head needs to be modified. To ensure a fair comparison with other models, the classification head was modified to a fully connected layer with 1024 neurons.

(2) Source domain pre-training (Teacher–Student self-distillation)

Since SDM uses the exponential average to update the Teacher model, it needs to store the historical weights of the Teacher model and update both the Teacher model and its weights with the parameters of the Student model at the end of each training round. To enhance model training, SDM introduces a round threshold to regulate the composition of the loss function. Initially, when training time is limited, random initialization of the model’s classification head can lead to poor classification results. Using the cosine alignment loss can cause learning difficulties. However, as the model progresses in its learning, gradually increasing the weight assigned to the cosine alignment loss can enhance the model’s generalization capabilities. This includes core loss calculations, gradient updates, and exponential average updates.

The cosine similarity loss function is used to determine whether the two vectors of the input are similar. It is commonly used for nonlinear word vector learning and semi-supervised learning. For batch data with N samples, D(a,b,y), a and b represent the two vectors entered, and y represents the real category labels, which belong to {1, −1}, representing similar and dissimilar, respectively. The loss corresponding to the ith sample is as follows: $$l_{i}=\left\{\begin{array}{cc} 1-\cos\left(a_{i},b_{i}\right) & if{\;y}_{i}=1\\ \max\left(0,\cos\left(a_{i},b_{i}\right)-margin\right) & if{\;y}_{i}=-1 \end{array}\right.$$
where the label ${\;y}_{i}=-1$ and $\cos\left(a_{i},b_{i}\right)<margin$, $l_{i}=0$. In this case, the input samples are not similar and the $\cos\left(a_{i},b_{i}\right)$ is relatively small, which is an easy-to-classify sample and is not included in the loss.

When the label $y_{i}=-1$ and $\cos\left(a_{i},b_{i}\right)>margin$, $l_{i}=\cos\left(a_{i},b_{i}\right)-margin$.

When the label $y_{i}=1$, $l_{i}=1-\cos\left(a_{i},b_{i}\right)$. In particular, when the angle between $a_{i}$ and $b_{i}$ is 0, $l_{i}=0$.

(3) Target domain fine-tuning (MixUp hybrid training)

In each batch, including image A, image B, and their corresponding labels, random numbers are generated according to the Beta distribution, α=0.5,β=α [25]. Subsequently, these random numbers are distributed evenly. In a situation where a random number falls within a certain range (e.g., 30% probability), the mixed images are utilized and the mixed classification loss is calculated. On the other hand, if the random number falls within another range (e.g., 70% probability), only image A is used for training. 

(4) Target domain test

To determine the Top1 classification accuracy, SDM categorizes the remaining data in the target domain.

### 3.2. Experimental Results and Analysis

In this paper, the accuracy average value is used as the evaluation index of the model to measure the classification results of six small samples across domains. Ablation study is a common experimental method in machine learning. By performing ablation experiments, it is possible to examine the importance of a part of the model, verify the extent to which relevant features affect the model results, and evaluate the contribution of different components, parameters, or algorithms to the model effect.

(1)Ablation experiments

To demonstrate the efficacy of the methods proposed in self-distillation and MixUp, the SDM model was initially validated through ablation experiments. These experiments involved removing the Self-Distillation module and MixUp module individually. The experimental outcomes are summarized in Table 1.

As shown in Table 1, when the self-distillation module is removed from SDM, the model’s average accuracy rate is 86.9%, which is 10.8% higher than that of the benchmark model. When the MixUp module is removed from SDM, the model’s average accuracy rate is 85.0%, which is 8.9% higher than that of the benchmark model. When SDM uses both self-distillation and MixUp, the model’s average accuracy rate is the highest, which is 2.1% and 4.0% higher than that of the models without self-distillation and MixUp, respectively.

(2)Comparison of the results

Using the Office31 and OfficeHome datasets, SDM conducted a series of experiments for the cross-domain small sample (supervised domain adaptation) task. The experimental results are summarized in Table 2 and Table 3.

Table 2 and Table 3 show that SDM has achieved the optimal results on the Office31 dataset, with an average accuracy that is 12.9% higher than the benchmark model (Resnet-50 pre-trained only in the source domain) and 1.0% higher than the CTL model proposed in 2023.

SDM achieved an impressive average accuracy of 52.7% on the OfficeHome dataset, outperforming the optimal model on select tasks. However, the Alexnet model, due to its simplicity and lack of residual connections like ResNet, experiences difficulties in gradient updates and is prone to instability, ultimately limiting its performance. We also tested the impact of utilizing the more advanced Resnet-50 model as our backbone and observed a significant 16.0% improvement in accuracy compared to Alexnet. This finding aligns with previous research that highlights the accuracy gap between Alexnet and Resnet-50 on ImageNet datasets [26,27,28]. Specifically, Alexnet has no residual structure and is weaker than ResNet for more complex distributions. However, it is also worth pointing out that the use of VIT, which has a stronger fitting ability and more complex model, has a poor effect, mainly because VIT’s fitting ability is too strong so it overfits on a small-sample dataset, which cannot solve the problem of fewer samples.

As shown in Table 2, SDM achieved the highest score in the “Avg” (average) category on the Office31 dataset, with significant advantages in the D→W and W→D tasks. Overall, the SDM model performed best among these six tasks. As shown in Table 3, SDM performs best on average across all tasks on the OfficeHome dataset, achieving the highest performance in domain adaptation tasks.

(3)Results analysis

To provide a more intuitive understanding of the effectiveness of the SDM method, this paper performs t-SNE analysis on three models: the benchmark model, self-distillation model, and fine-tuning model. The amazon subset of the Office31 dataset serves as the source domain, and the webcam subset serves as the target domain. Eight classifications are randomly chosen, with 10 samples selected for each classification. Each type is distinguished by a unique color for visual classification purposes. Subsequently, the feature vector of each image is computed using the trained backbone. We employ the T-SNE dimensionality reduction technique to project the high-dimensional feature vectors onto a 2-dimensional plane, which are then plotted onto the image.

After pre-training on the amazon domain, the t-SNE analysis of the pure ResNet model is shown in Figure 5, where different colors represent different categories.

After pre-training on the amazon domain, the t-SNE analysis of the Teacher–Student model is shown in Figure 6.

It is evident that the self-distillation approach, which is based on the Teacher–Student method, exhibits a smaller intra-class distance in the source domain compared to the simple ResNet after the pre-training phase. Furthermore, it demonstrates comparable generalizability in the target domain.

Furthermore, Figure 6 reveals that the pre-trained relationship not only diminishes the intra-class distance in the source domain but also enhances the inter-class distance. Even more impressively, the extracted features can be effortlessly classified through a linear classifier. Moreover, it is evident that despite the model’s lack of exposure to target domain data, samples from the target domain are still accessible. This ensures a relatively small relative intra-class distance and a certain aggregation phenomenon. This observation highlights the significance of the Teacher–Student architecture. It is because increasing the separability of feature vectors in the feature space not only elevates the inter-class distance but also diminishes the intra-class distance.

To reflect the effect of fine-tuning on domain adaptation, a visual analysis of the model after fine-tuning of the target domain using the MixUp method is shown in Figure 7.

After fine-tuning, it is evident from Figure 7 that all classes are noticeably positioned in distinct regions. The model primarily expands the inter-class distance in the target domain, thereby enhancing the model’s separability. However, the SDM algorithm still has some limitations. The detection effect of SDM depends on the quality of the input image. If there are many noise points in the input image, it will affect the detection effect.

## 4. Conclusions

To address the challenges of few-shot sizes in practical training deployment, this paper introduces a meta-learning-based approach to mitigate domain offset issues. Specifically, we propose two techniques: meta-learning fine-tuning and pre-training meta-learning. Additionally, we introduce a self-distillation and mixing (SDM) method, borrowing insights from the Teacher–Student framework. This approach transfers knowledge from the source domain to the target domain, leveraging self-distillation and hybrid data enhancement techniques. Experimental results demonstrate that the SDM method offers superior performance in terms of training time and accuracy, effectively transferring knowledge from the source domain to the target domain, even with a limited number of target domain samples. The SDM model can be effectively applied in sparse sample environments, such as in scenarios for detecting welding defects.

## Figures and Tables

**Figure 1 sensors-24-01939-f001:**
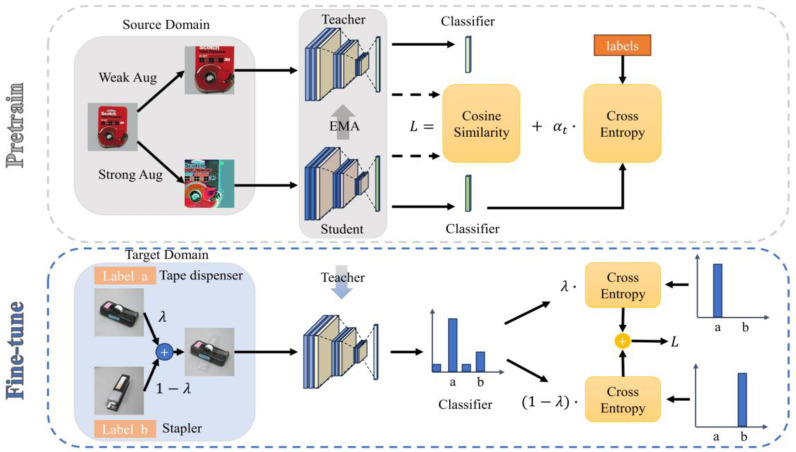
Schematic representation of the SDM.

**Figure 2 sensors-24-01939-f002:**
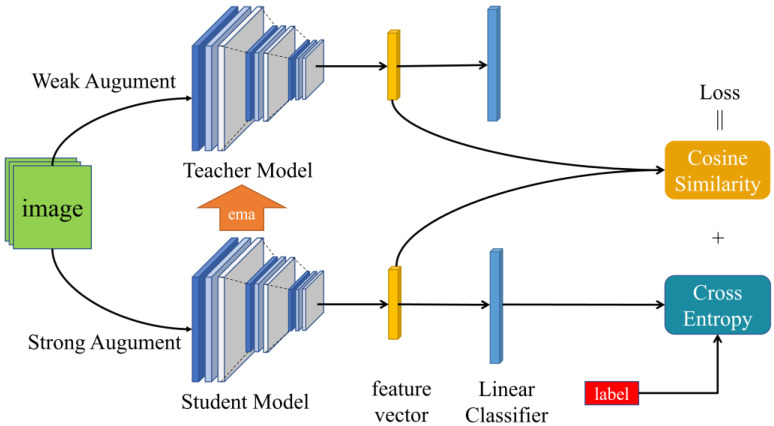
SDM model diagram in the pre-training stage.

**Figure 3 sensors-24-01939-f003:**
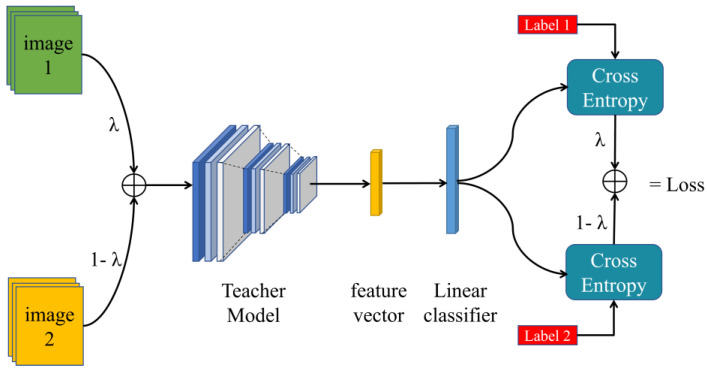
SDM model schematic in the fine-tuning stage.

**Figure 4 sensors-24-01939-f004:**
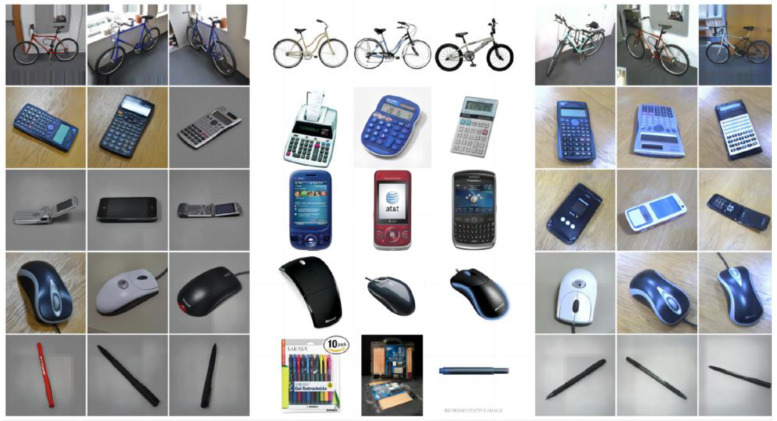
Example of office31 Sample (**left**: DSLR; **middle**: amazon; **right**: webcam).

**Figure 5 sensors-24-01939-f005:**
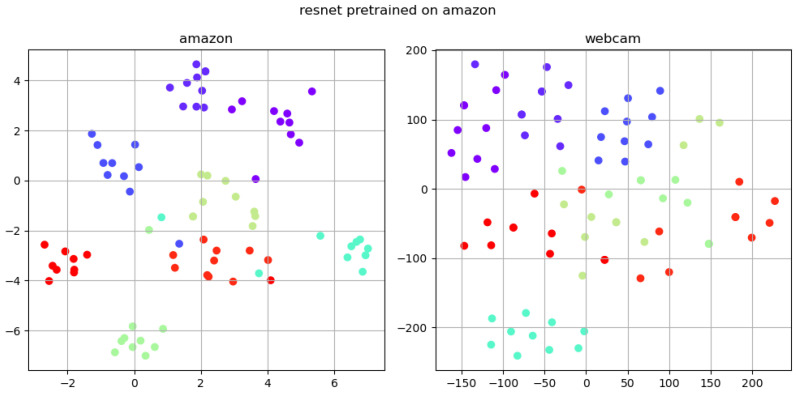
Resnet The t-SNE analysis of the pre-training results.

**Figure 6 sensors-24-01939-f006:**
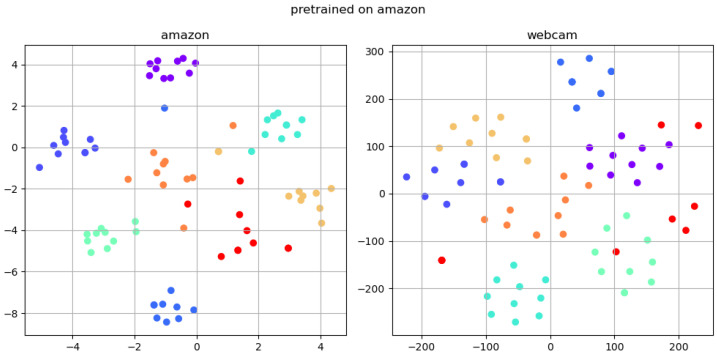
The t-SNE analysis after pre-training.

**Figure 7 sensors-24-01939-f007:**
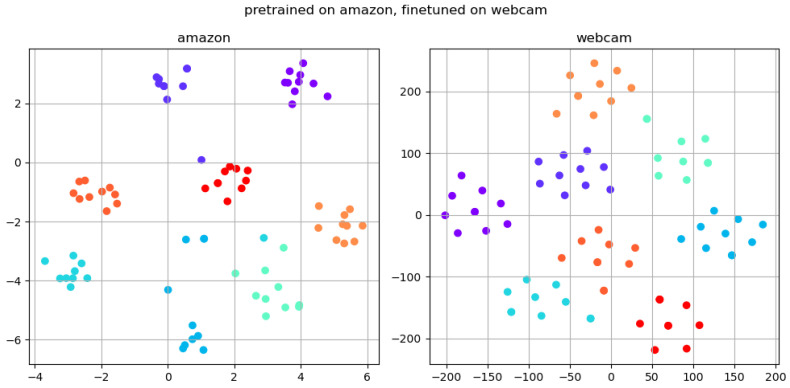
t-SNE analysis after pre-training + fine-tuning.

**Table 1 sensors-24-01939-t001:** Average classification accuracy (%) on the 31 Office31 dataset.

Method	A→D	A→W	D→A	D→W	W→A	W→D	Avg
ResNet-50	68.9	68.4	62.5	96.7	60.7	99.3	76.1
w/o Self- Distillation	93.0	90.3	73.0	97.1	70.2	98.0	86.9
w/o Mixup	87.7	86.2	68.0	98.3	69.8	**100.0**	85.0
SDM(Ours)	**93.3**	**91.2**	**74.5**	**99.1**	**76.6**	99.3	**89.0**

Note: ResNet-50 refers to pre-training only in the source domain. w/o self-distillation indicates that the source domain pre-training stage does not employ the self-distillation method. Additionally, w/o MixUp signifies that the target domain fine-tuning stage does not utilize MixUp for training directly. A: amazon, D: DSLR, W: webcam. The parts highlighted in bold are the maximum values.

**Table 2 sensors-24-01939-t002:** Average classification accuracy (%) of the different models on the 31 classes of the Office31 dataset.

Method	A→D	A→W	D→A	D→W	W→A	W→D	Avg
ResNet-50	68.9	68.4	62.5	96.7	60.7	99.3	76.1
SDADT [11,12,13,14,15,16,17,18]	86.1	82.7	66.2	95.7	65.0	97.6	82.2
CCSA [7,8,9,10,11]	89.0	88.2	71.8	96.4	72.1	97.6	85.8
FADA [8,9,10,11,12]	88.2	88.1	68.1	96.4	71.1	97.5	84.9
d-SNE [9,10,11,12,13]	91.4	90.1	71.7	97.5	71.1	97.1	86.5
DAG-LDA [19,20,21]	85.9	87.8	66.5	97.9	64.2	99.5	83.6
MF [20,21,22]	90.0	87.3	72.1	97.2	72.4	96.5	85.9
So-HoT [21,22,23]	86.3	84.5	66.5	95.5	65.7	97.5	82.7
CTL [10,11,12,13,14]	92.1	89.4	74.4	98.3	74.2	**99.4**	88.0
SDM(Ours)	**93.3**	**91.2**	**74.5**	**99.1**	**76.6**	99.3	**89.0**

Note: ResNet-50 is only pre-trained in the source domain, and the best performance is highlighted in boldface.

**Table 3 sensors-24-01939-t003:** Average classification accuracy (%) of the different methods on the OfficeHome dataset.

DA Tasks	Alexnet	CCSA	d-SNE	DAG-LDA	CTL	SDM (Ours)	SDM (Ours) *
Ar→Ca	28.2	41.3	40.3	40.8	**42.9**	41.2	**55.7**
Ar→Pr	39.5	57.3	58.2	55.3	**61.7**	57.1	**76.5**
Ar→Rw	51.4	59.2	57.1	57.4	**62.5**	60.5	**78.3**
Ca→Ar	32.0	42.5	41.5	41.3	**43.8**	41.9	**61.1**
Ca→Pr	44.9	**59.1**	56.3	57.2	57.6	**60.1**	**72.2**
Ca→Rw	47.1	59.1	58.2	58.4	59.0	58.5	**71.1**
Pr→Ar	30.2	40.0	40.2	**42.2**	41.5	41.2	**63.3**
Pr→Ca	28.7	44.1	43.9	44.0	45.1	**45.6**	**54.8**
Pr→Rw	53.6	58.9	58.4	59.2	**60.4**	59.8	**77.7**
Rw→Ar	43.4	46.4	46.2	48.2	50.6	**51.7**	**70.9**
Rw→Ca	33.9	45.2	46.6	46.7	**48.0**	46.6	**60.3**
Rw→Pr	60.6	66.9	68.2	67.4	**71.3**	68.1	**82.0**
Avg	41.1	51.7	51.3	51.5	**53.7**	52.7	**68.7**

Note: * Based on Resnet-50, other methods based on AlexNet, best performance highlighted in boldface.

## Data Availability

Data are contained within the article.

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
