# Peer review of "Research on a Cross-Domain Few-Shot Adaptive Classification Algorithm Based on Knowledge Distillation Technology"

_sensors, 2024, doi:10.3390/s24061939_

Round 1
Reviewer 1 Report
Comments and Suggestions for Authors
To improve the quality of the manuscript, major revision is required. The following comments should be considered into the account.
1. Authors should describe, how might the insights gained from this study inform the development of more robust and efficient deep learning algorithms for classification tasks with limited samples and domain offset?
2. In this current study, how the performance of the SDM method compare to nine classical models in terms of training time and accuracy? Were any statistical tests conducted to validate the significance of the results?
3. Authors should add some motivated the choice of self-distillation and hybrid data augmentation as key components of the SDM method?
4. How does the proposed SDM approach address these challenges, particularly in the context of few-shot learning problems?
5. What are the potential practical applications or implications of the SDM approach in real-world scenarios?
6. An important issue of the considered model is to get faster convergence speed. Which factors will influence the convergence speed?
7. What’s the limitation of your proposed model? Are there other ways that the results can be further improved? One or two remarks should be given to discuss it in detail.
Comments on the Quality of English LanguageNeed to check grammatical errors.
Author Response
Thanks to the reviewer's suggestion. Please refer to the attachment for specific modification

Reviewer 2 Report
Comments and Suggestions for Authors
This paper introduced a novel self-distillation and mixing model based on the Teacher-Student learning framework for the task of object detection on image data. The authors try to alleviate the cross-domain learning problem as the distribution between the source domain and target domain is different which is widely existing in the real-world applications. This work describes the proposed algorithm properly and explains the training process in details.
Here are few highlights of this work:
-
The strong augmentation, the weak augmentation and the loss function design in the proposed model improve the learning robustness.
-
The introduced the fine-tuning technique by randomly sample the training data with the Beta distribution can improve the feature representation ability toward the target domain.
-
The abundant experiment results presented by the authors.
Moreover, here are few issues that the authors can improve the work into a better version:
1, The format of the paper requires improvements. For example, the titles of Figures and Tables are not in good quality considering this journal’s requirements. The copy and paste of the algorithm in Figure 4 needs improvement.
2, The meaning of several symbols are missing. For example. The symbol “num” does not have a definition in the oss function of SDM (as in Page 4). The meaning of “A->D”, “A->W” is not clear. I guess it follows the description in the paragraph 2 of Page 6.
3, Regarding the performance of the proposed model, especially follow the result of Table 3, it seems the based pre-trained model like AlexNet and Resnet-50 have big impact to the final performance. The authors may want to give more deep studies to understand the root cause.
Comments on the Quality of English Language
The English writing of this paper has moderate quality. I didn't find any difficulty in reading the paper. However, the format does need improvements, especially the Figures and Tables.
Author Response
Thanks to the reviewer's suggestion. Please refer to the attachment for specific modification.

Reviewer 3 Report
Comments and Suggestions for Authors
The paper introduces a compelling method for cross-domain few-shot adaptive classification employing knowledge distillation technology. However, suggest undergoing a significant revision to address several crucial aspects:
How were the two NVIDIA 2080Ti GPUs utilized in the experimental environment? Were they utilized for parallel processing or other tasks?
Could you provide more details about the SFAN (Serverless Auto-Scale) function and its relevance to the experimental setup?
Regarding the adjustment of the dataset, what specific techniques were employed for weak and strong image enhancement during model training?
How were the hyperparameters determined for the Self-Distillation and MixUp modules, and were there any sensitivity analyses conducted on these parameters?
Provide more insights into how the cosine alignment loss was implemented and how its weight was gradually increased during training.
In the MixUp hybrid training, how were the random numbers generated according to the Beta distribution, and what specific parameters were used?
Explain the reason of conducting ablation experiments and how the results from these experiments contribute to the validation of the proposed methods.
In the t-SNE analysis, how were the feature vectors extracted from the benchmark model, self-distillation model, and fine-tuning model, and how were they visualized?
The manuscript can better establish its relevance within the broader field of knowledge distillation by integrating the below study, showcasing its significance across diverse domains. This recommendation aims to bolster the manuscript's theoretical underpinning and underscore the importance of its contributions.
- Aligning Distillation For Cold-Start Item Recommendation. https://doi.org/10.1145/3539618.3591732
Author Response

(The authors gave the same response as above.)

Round 2
Reviewer 1 Report
Comments and Suggestions for Authors
The authors answered all the comments. Thus it can be suitable for publication
Comments on the Quality of English LanguageAuthors need to check the reference format.
Reviewer 2 Report
Comments and Suggestions for Authors
The improved version shows better quality of writing, better quality of images and tables. I think the paper is in good shape now.
Comments on the Quality of English LanguageThe quality of the English is good. I didn't find any difficulty to read it.
Reviewer 3 Report
Comments and Suggestions for Authors
In my opinion, the revised version can be acceptable for publication.